# Forest Tree Associated Bacterial Diffusible and Volatile Organic Compounds against Various Phytopathogenic Fungi

**DOI:** 10.3390/microorganisms8040590

**Published:** 2020-04-18

**Authors:** Wei-Liang Kong, Pu-Sheng Li, Xiao-Qin Wu, Tian-Yu Wu, Xiao-Rui Sun

**Affiliations:** 1Co-Innovation Center for Sustainable Forestry in Southern China, College of Forestry, Nanjing Forestry University, Nanjing 210037, China; k3170100077@njfu.edu.cn (W.-L.K.); lps1996127@163.com (P.-S.L.); skyuwu@126.com (T.-Y.W.); taylorrui240@njfu.edu.cn (X.-R.S.); 2Jiangsu Key Laboratory for Prevention and Management of Invasive Species, Nanjing Forestry University, Nanjing 210037, China

**Keywords:** diffusible substances, volatile compounds, phytopathogenic fungi, *Pseudomonas* sp.

## Abstract

Plant growth-promoting rhizobacteria (PGPR) can potentially be used as an alternative strategy to control plant diseases. In this study, strain ST–TJ4 isolated from the rhizosphere soil of a healthy poplar was found to have a strong antifungal activity against 11 phytopathogenic fungi in agriculture and forestry. Strain ST–TJ4 was identified as *Pseudomonas* sp. based on 16S rRNA-encoding gene sequences. The bacterium can produce siderophores, cellulase, and protease, and has genes involved in the synthesis of phenazine, 1–phenazinecarboxylic acid, pyrrolnitrin, and hydrogen cyanide. Additionally, the volatile compounds released by strain ST–TJ4 can inhibit the mycelial growth of plant pathogenic fungi more than diffusible substances can. Based on volatile compound profiles of strain ST–TJ4 obtained from headspace collection and GC–MS/MS analysis, 1-undecene was identified. In summary, the results suggested that *Pseudomonas* sp. ST–TJ4 can be used as a biocontrol agent for various plant diseases caused by phytopathogenic fungi.

## 1. Introduction

Severe crop loss remains inevitable due to plant diseases, particularly those caused by pathogenic fungi, which are responsible for an estimated 70–80% of plant diseases [1]. Phytopathogenic fungi reduce both crop yield and quality. They are major restraints to sustainable agriculture production, especially in intensive cropping systems [2]. Synthetic fungicides have been used extensively to control diseases caused by these pathogens [3]. However, these chemicals may lead to toxic residues in treated products [4,5]. Synthetic pesticides can also pollute the environment due to their slow biodegradation and can induce resistance or reduce the susceptibility of pathogenic fungi. Furthermore, that the species and numbers of fungal phytopathogens in the rhizosphere change with environmental conditions and evolution increases the difficulty of controlling plant diseases [6]. Facing the severe threat to global crop security caused by plant diseases, it is important to develop environmentally friendly and highly effective fungicides against plant pathogens. 

Compared with application of synthetic chemical fungicides, the use of microorganisms and their metabolites is a promising and environmentally friendly alternative for the prevention and control of plant diseases [7]. Biological control agents, such as *Bacillus*, *Pseudomonas*, *Burkholderia*, and *Paenibacillus* spp. play important roles in inhibiting pathogens [8,9,10,11]. Bacteriocin, lipopeptide and polyketide are active substances produced by *Bacillus* species against pathogens [12]. The siderophore-mediated competition for iron gives beneficial microbes a competitive advantage in suppressing the proliferation and root colonization of plant pathogens [13]. 

In addition, microbial volatile organic compounds (VOCs) have attracted more attention because they can spread over long distances, mediating indirect contact interactions between organisms. VOCs at low concentrations can be sensed, so they can directly inhibit the growth of pathogenic fungi and induce systemic resistance in plants [14,15]. The VOCs produced by *Bacillus* can inhibit mycelial growth of *Alternaria solani* and *Botrytis cinerea*, which cause early blight of potatoes and grey mold of a broad range of hosts, respectively [16]. They can also control *Ceratocystis fimbriata* in postharvest sweet potatoes [17]. It is suggested that the application of microbial volatiles has a great potential in plant diseases. 

Current research on biocontrol bacteria is mainly focused on crop applications and rarely on forest diseases. In preliminary analyses prior to this study, we screened a bacterial strain with high siderophore production from the poplar rhizosphere; however, it remains unclear whether this strain has antagonistic effects on fungi causing forest diseases or produces additional antagonistic substances. Therefore, in this study, we identified diffusible substances and VOCs produced by this strain to determine whether they may be applied as antagonistic substances. We also determined the taxonomic placement of this strain via morphological identification and molecular biology. The results of this study may provide a new and effective antagonistic bacterial strain for the biological control of forest fungal diseases, and lay a foundation for molecular analyses of the bacteriostatic mechanism of this strain.

## 2. Materials and Methods

### 2.1. Bacterial and Fungal Strains

In this study, bacterial strain ST–TJ4 was isolated from poplar rhizosphere soil in 2018 at Tianjin, China. Strain ST–TJ4 was stored at −80 °C in Luria–Bertani (LB) medium with 50% (*v*/*v*) glycerol for long–term use. The fungal plant pathogens studied in this study were *Botryosphaeria berengeriana* (causes apple ring rot); *Colletotrichum tropicale* (causes *Ficus binnendijkii* var. variegate anthracnose); *Cytospora chrysosperma*, *Fusicoccus aesculi*, and *Phomopsis ricinella* (cause poplar canker); *Fusarium oxysporum* (causes cotton wilt); *Fusarium graminearum* (causes wheat head blight); *Phytophthora cinnamomi* (causes cedar root rot); *Pestalotiopsis versicolor* (causes tea round spot); *Rhizoctonia solani* (causes pine seedling damping-off) and *Sphaeropsis sapinea* (causes pine shoot blight). These fungal plant pathogens were preserved in the Forest Pathology Laboratory of Nanjing Forestry University. The isolates were maintained on Potato dextrose agar (PDA) plates at 25 °C.

### 2.2. In Vitro Antifungal Activity

According to the method of Lim et al. [18], we used a 0.6 cm hole punch to take the plugs containing mycelia from the aforementioned 11 phytopathogenic fungi cultured for 7 days and put the plugs in the center of PDA culture medium individually. Then, we dipped a sterile loop into the overnight culture of ST–TJ4 bacterial suspension, and streaked 2.5 cm from one side of the plug. After 5–7 days, the width of the inhibition zone between the bacterial colony and fungal pathogen was measured. Fungi that were not inoculated with bacteria served as a control. The plant pathogenic fungus inhibition rate was calculated as inhibition rate (%) = (*C_d_* – *T_d_*) × 100%/*C_d_*, where *C_d_* is the radial mycelial growth in control, and *T_d_* is the radial growth of the fungal pathogens in treatment (dual culture). Each treatment had four replicates. The experiment was also repeated twice.

### 2.3. Analysis of the Antagonistic Substances In Vitro

According to the previous methods [19,20,21,22], Chrome Azurol S (CAS) agar plates, carboxyl methyl cellulose (CMC) agar plates, skim milk powder (SMP) agar plates, colloidal chitin agar plates, and Pachyman solid medium were made to detect the production of siderophore, cellulase, protease, chitinase, and β–1,3 glucanase, respectively. The single colony of ST–TJ4 picked with a toothpick was used to inoculate the aforementioned media, which were incubated in the dark at 28 °C. After three days, the transparent circle around the colony was observed.

### 2.4. Detection of Genes Encoding Antibiotics and HCN in ST–TJ4

Total DNA was isolated from ST–TJ4 cells by the cetyltrimethylammonium bromide (CTAB) method [23]. Then, polymerase chain reaction (PCR) assays were used to detect the *phzCD*, *phz*, and *prnC* genes according to protocols described in Raaijmakers et al. [24] and Hu et al. [25]. The *hcnAB* gene was detected as previously described [26]. The primers used in the experiment are listed in Table 1.

### 2.5. Antifungal Activity of VOCs of Strain ST–TJ4

The method using two sealed base plates was employed to test the antifungal activity of VOCs from strain ST–TJ4 [16]. One base plate contained 20 mL LB, and the other contained 20 mL PDA. The LB medium was coated with 100 μL of ST–TJ4 suspension, and a 0.6 cm-diameter plug of phytopathogenic fungi was placed in the center of PDA plates. Next, the two base plates were sealed with Parafilm and cultured in a fungal incubator for 5–7 d at 25 °C. Every experiment was repeated three times. The inhibition rate of each plant pathogenic fungus was calculated as inhibition rate (%) = (*C_d_* – *T_d_*) × 100%/*C_d_*, where *C_d_* is the fungal colony diameter on the control PDA base plate, and *T_d_* is the fungal colony diameter on the treatment PDA base plate.

### 2.6. Analysis of Volatile Organic Compounds (VOCs)

Volatile organic compounds from LB liquid medium inoculated with strain ST–TJ4 were analyzed using solid phase microextraction–gas chromatography–mass spectrometry (SPME–GC–MS/MS) with a SPME fiber assembly (CAR/PDMS; Supelco, Bellefonte, PA, USA). The single colony was inserted into a 200 mL flask containing 50 mL liquid LB medium and fermented on a shaker at 180 rpm at 28 °C for 2 days. To prevent volatile organic compounds from escaping, the flask was sealed with tin foil. The LB liquid medium without inoculation of bacterium was used as the control. 

In this experiment, the 65 µm Polydimethylsiloxane/Divinylbenzene (PDMS/DVB) fiber was selected for the determination of bacterial VOCs. The extraction head must be aged when it is used for the first time. In this experiment, the aging temperature of the extraction head was 250 °C, and the time was 30 min. The extraction fiber was put into the SPME holder, the SPME needle was inserted into the gasification chamber of gas chromatography, and the plunger was pressed to push the fiber out the fiber attachment tubing (needle) for aging. At the end of aging, the fiber was retracted into the fiber attachment tubing and then the SPME needle was pulled out. The cultured bacterial sample was shaken and put in a water bath at 40 °C for the volatilization of the gas. The needle of the aged SPME was inserted through the tin foil to press down the plunger to make the fiber protrude the tubing so that the fiber was exposed to the upper air of the bacterial fermentation broth to absorb and extract VOCs for 30 min, during which the flask was shaken by hand every five minutes, so that the gas can be more easily released from the liquid surface, but the shaking should be gentle to avoid the fermentation liquid contacting the fiber, contaminating the fiber and affecting the extraction effect. After the extraction, the fiber was quickly retracted and the needle was pulled out, and immediately inserted into the gasification chamber of gas chromatography (Agilent 7000B, Palo Alto, CA, USA), the fiber was pushed out, and the sample was analyzed at high temperature in the gasification chamber for 3 min, then the fiber was retracted to the stainless steel casing and pulled out for collecting next sample.

GC–MS settings: using Rtx–5 quartz capillary column, carrier gas was He; injector port temperature 230 °C; initial temperature was 40 °C, which was maintained for 3 min, heating up to 95 °C at 10 °C /min, then to 230 °C at 3 ℃/min , and 230 °C was maintained for 5 min; ion source—EI source; electron energy—70eV; spectrum retrieval carried out using Nist 05 and Nist 05s spectrum library.

### 2.7. Identification of Strain ST-TJ4

The colony morphology and physiological and biochemical characteristics were analyzed according to the Handbook of Common bacterial Identification. Molecular identification: CTAB method was used to extract and purify bacterial genomic DNA, and universal primers were 27F (5’–AGAGTTGATCATGGCTCAG–3’) and 1492R (5’–TACGGYTACCTTGTACGACTT–3’) to amplify 16S rRNA–encoding gene [23]. After PCR products were ligated and transformed, the positive clones were selected and sent to Jie Li Biotechnology Co., Ltd. for sequencing, and the sequence data of Blast and GenBank were compared and analyzed. The phylogenetic tree was constructed by screening and choosing the sequences of closely related species using MEGA 7.0 software [27].

### 2.8. Statistical Analysis

The data were performed by analysis of variance and Duncan multiple comparison with SPSS 21.0 software (IBM Inc., Armonk, NY, USA), and the standard errors of all mean values were calculated (*p* < 0.01). Graphs were drawn using GraphPad Prism 8.0 (GraphPad Software, Inc., San Diego, CA, USA).

## 3. Results

### 3.1. Antagonistic Effects of Diffusible Substances Produced by ST–TJ4

Strain ST–TJ4 inhibited the mycelial growth of all fungal pathogens tested to different degrees. ST–TJ4 significantly inhibited the mycelial growth of *B*. *berengeriana*, *F*. *graminearum*, *F*. *aesculi*, *F*. *oxysporum*, *Pe*. *versicolor*, *Ph*. *cinnamomi*, *P*. *ricinella* and *S*. *sapinea*, but only weakly affected that of *C*. *chrysosperma*, *Co*. *tropicale* and *R*. *solani* (Figure 1; Table 2).

### 3.2. Siderophore and cell wall Degradation Enzyme Activities of ST–TJ4

To examine the antimicrobial substances produced by ST–TJ4, including cellulase, chitinase, protease, and glucanase, ST–TJ4 was inoculated on PDA plates containing cellulose, colloidal chitin, SMP, or β–glucan. After 3 days, an obvious hydrolysis zone appeared on the plates containing cellulose and SMP, but not on those containing chitin or β–glucan (Figure 2). These results indicated that ST–TJ4 has strong siderophores, cellulase, and proteolytic activities, but no chitinolytic or glucanolytic activities.

### 3.3. Detection of Antibiotic and HCN Encoding Genes in ST-TJ4

We tested strain ST–TJ4 for the presence of operons for the biosynthesis of the antibiotics, phenazine–1–carboxylic acid (PCA), phenazine (PHZ), pyrrolnitrin (PRN), and hydrogen cyanide (HCN) by PCR using specific primers. A fragment of the predicted size for each of these compounds was observed from the DNA of the reference *Pseudomonas* strains, which produces these compounds. Fragments of the predicted sizes for PCA (~1150 bp), PHZ (~1408 bp), PRN (~786 bp), and HCN (~570 bp) were amplified from strain ST–TJ4 DNA (Figure 3).

### 3.4. The Antifungal Spectrum of ST-TJ4 VOCs

The two–sealed–base–plates method was used to determine the antifungal spectrum of ST–TJ4 VOCs (Appendix A). Strain ST–TJ4 VOCs showed significant antifungal activities against 11 plant pathogenic fungi. Although the effects against *Pe*. *versicolor*, *F*. *graminearum*, and *F. oxysporum* were weak, the mycelial growth and pigment secretion of the other 8 pathogens were significantly inhibited by 40.4–91.4% (Figure 4; Table 2). Moreover, comparing the antagonistic effects of VOCs and diffusible substances, the inhibitory effect of VOCs on all pathogens exceeded those of diffusible substances, suggesting that volatile compounds predominate in biological control by ST–TJ4.

### 3.5. GC-MS/MS Analysis of VOCs Produced by ST–TJ4

Volatiles from strain ST–TJ4 were collected in a SPME syringe and analyzed with a GC–MS/MS system. The same volatiles in the LB medium and substances with relative contents less than 0.5% were filtered out. This revealed very clear separation between the control and strain ST–TJ4, as indicated in Figure 5. Five different VOCs emitted by strain ST–TJ4 were identified. The most abundant volatile produced by strain ST–TJ4 was 1–undecene, which had a large peak (75.97%) at a retention time (RT) of 8.65 minutes (Table 3). The other compounds were l–Ala–l–Ala–l–Ala (1.42%), octamethylcyclotetrasiloxane (1.14%), 4-hydroxybenzoic acid (1.06%), and phosphonoacetic acid (0.54%).

### 3.6. Identification of Strain ST–TJ4

Colonies of ST–TJ4 on LB plates appeared thin, flat, orange, opaque, round with smooth edges, and approximately 1.2–3 mm in diameter (Figure 6A). The oxygen demand test showed that ST–TJ4 is an aerobic bacterium. It grows well outside the cover glass, but hardly grows under the cover glass (Figure 6B). Gram staining showed that ST–TJ4 is a Gram-negative bacterium (Figure 6C). Analysis of 16S rRNA–encoding gene sequence showed that strain ST–TJ4 formed a monophyletic group with *Pseudomonas* species, sharing 99% sequence similarity (Figure 7). Based on the morphological characteristics and phylogenetic analysis, strain ST-TJ4 was identified as *Pseudomonas* sp.

## 4. Discussion

It is an urgent problem to develop environment-friendly products instead of chemicals to control plant diseases [28]. Plant pathologists and microbiologists believe that the use of beneficial microorganisms and their metabolites in the future crop production practice is safe, reasonable and one of the most promising methods [29]. Therefore, we tried to isolate biocontrol bacteria with strong spectral antagonistic activities against plant pathogenic fungi. We found that strain ST–TJ4 had strong inhibitory effect on 11 pathogenic fungi and oomycetes from branches, leaves and roots of forest trees. This suggested that strain ST–TJ4 has a wide fungicide spectrum and could produce antifungals. Through morphological observation, physiological and biochemical experiments as well as molecular biological identification, the strain ST–TJ4 was identified as *Pseudomonas* sp.

Iron is one of the indispensable nutrients in all organisms. Although total iron is rich in the most soils, most iron exists either in chemical forms that cannot be used by microorganisms or imbedded deeply in mineral particles. Pathogens need to compete with other members of a microbial community for scarcely available iron before the host plant is infected. Therefore, iron acquisition is the basis of the complete virulence of many plant pathogens [30]. Many PGPR can produce siderophores that can specifically chelate Fe^3+^ under the condition of iron deficiency. Studies have shown that these siderophores can inhibit the growth of pathogens by depriving iron, so as to achieve the purpose of controlling plant diseases. Tortora et al. [31] have achieved good results in using catechol iron carriers produced by *Azospirillum brasiliensis* to control strawberry anthracnose caused by *Colletotrichum cuspidatum*. The siderophores produced by *Pseudomonas clove* BAF1 has a great effect on the spore germination and mycelial morphology of *Fusarium oxysporum* [32]. Strain ST–TJ4 was screened by siderophile production, and its strong chelating ability of Fe^3+^ is also expected to be one of the reasons for its inhibition of pathogen growth. 

Biocontrol bacteria can inhibit the growth of other microorganisms by producing cell wall degrading enzymes and secondary metabolites. It is well known that fungal cell walls play an extremely important role in cell division, maintaining the normal growth of hyphae and coping with environmental stress [33,34]. Moreover, maintaining the integrity of fungal cell wall is the key to host infection [35]. The cell wall of true fungi is mainly composed of chitin, glucan and protein [36,37,38]. Once these components are destroyed, the mycelial morphology and biological function of fungi will be threatened. In this study, ST–TJ4 showed strong cellulase and protease hydrolysis activity and could degrade the cell wall of oomycetes.

Antibiotics produced by beneficial microorganisms have direct contact killing effect on pathogenic bacteria [39,40]. Therefore, antibiotics are not only one of the most studied mechanisms of biological control, but also a character to be considered when screening potential biocontrol agents (BCAs) [41,42]. Strain ST–TJ4 could change the color of the whole culture medium to orange after three days in LB culture, and the production of pigment suggested that the strain might produce metabolites. In order to test our hypothesis, we examined the antibiotic-related genes in ST–TJ4 with reference to the results of previous studies on *Pseudomonas* antibiotics. The PCR results suggested that ST–TJ4 contains genes for the biosynthesis of PCA and PRN. Phenazine type of antibiotics, such as PCA, are particularly active against oomycetes [43,44]. Park et al. [45] demonstrated that PRN produced by *P*. *chlororaphis* O6 plays a pivotal role in inhibiting *Phytophthora infestans*, which causes tomato late blight disease. In addition, genes related to the biosynthesis of HCN were also detected in strain ST–TJ4, and several recent studies reported that HCN may be involved in the effective control of wheat sheath blight by several *Pseudomonas* spp [46].

The volatile gas produced by biocontrol bacteria can inhibit the spore germination and mycelial growth of plant pathogens, and it has attracted wide attention [47]. Compared with non-volatile antimicrobial substances, volatile fungistatic substances have smaller molecules, are easier to permeate and spread in soil and atmosphere, and can kill pathogens in the environment more comprehensively [16]. Our study analyzed the VOCs produced by strain ST–TJ4 using SPME–GC–MS/MS and detected five compounds. Hunziker et al. [48] and Guevara-Avendaño et al. [49] reported that 1-undecene, which has strong antifungal activities, was the main active compound released by *Pseudomonas fluorescens*. Similarly, in our analysis, 1–undecene was the most abundant volatile in strain ST–TJ4. The antifungal activity of 1–undecene against *R*. *solani* AG–1(IA), which causes banded leaf and sheath blight (BLSB) of maize, has been reported [50]. The volatile methyl siloxane octamethylcyclotetrasiloxane (D4) has been reported to have antibacterial and antifungal activities [51] and was found in the present study. To our knowledge, this is the first report on the emission of 1–undecene and octamethylcyclotetrasiloxane by *Pseudomonas* sp., which has a broad fungistatic spectrum.

Interestingly, the diffusible substances produced by ST–TJ4 showed weak antagonistic effects against *C*. *chrysosperma*, *R*. *solani*, and *Co*. *tropicale*, whereas its volatiles almost completely inhibited the growth of these three pathogenic fungi. Because we used LB medium to culture bacteria in the two–sealed–base–plates method, while strain ST–TJ4 was streaked onto PDA plates in the dual–culture experiment. It is possible that the inoculation strategy led to different antagonistic effects; that is, the inhibition rate of spreading small cells may not have been as great as that of the spread cells exposed to larger surfaces of the medium, which would allow the single cells to obtain more nutrients and facilitate their rapid reproduction [52]. The results of Valentina et al. [53]. support our speculation. Only when a variety of biocontrol mechanisms are combined and complement each other can microorganisms play the greatest role in biological control.

To conclude, our results suggest that *Pseudomonas* sp. ST–TJ4 is a good biocontrol agent candidate, although it is unclear how effective this antagonist would be under the field conditions. Diffusible and volatile compounds produced by ST–TJ4 have the potential to be used in agriculture and forestry as direct contact biofungicides and biofumigants.

## Figures and Tables

**Figure 1 microorganisms-08-00590-f001:**
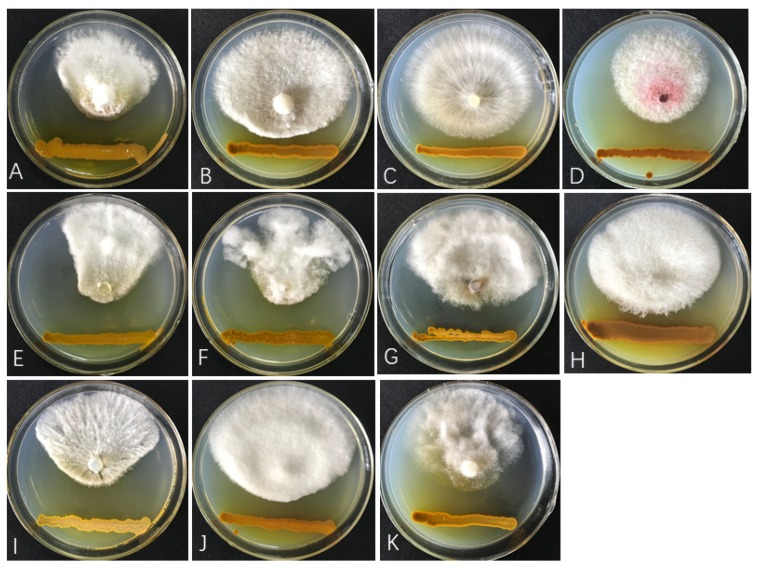
In vitro antifungal activity of ST-TJ4 against 11 phytopathogenic fungi in dual-culture assays. (**A**) *Pestalotiopsis versicolor*; (**B**) *Cytospora chrysosperma*; (**C**) *Rhizoctonia solani*; (**D**) *Fusarium graminearum*; (**E**) *Phomopsis ricinella*; (**F**) *Botryosphaeria berengeriana*; (**G**) *Fusicoccus aesculin*; (**H**) *Colletotrichum tropicale*; (**I**) *Sphaeropsis sapinea*; (**J**) *Fusarium oxysporum* and (**K**) *Phomopsis ricinella.*

**Figure 2 microorganisms-08-00590-f002:**
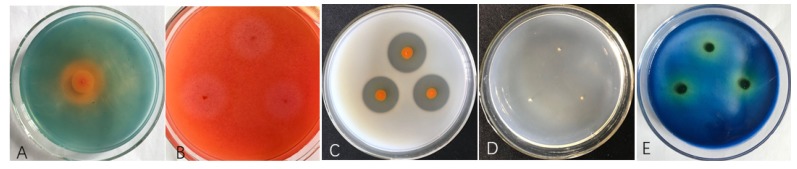
Fungal cell-wall-degrading enzymes produced by ST-TJ4 cells. (**A**) Siderophores on CAS plates; (**B**) cellulase activity on carboxyl methyl cellulose (CMC) plates; (**C**) protease activity on SMP plates; (**D**) chitinase activity on colloidal chitin agar plates and (**E**) glucanase activity on Pachyman solid medium supplemented with 6% aniline blue.

**Figure 3 microorganisms-08-00590-f003:**
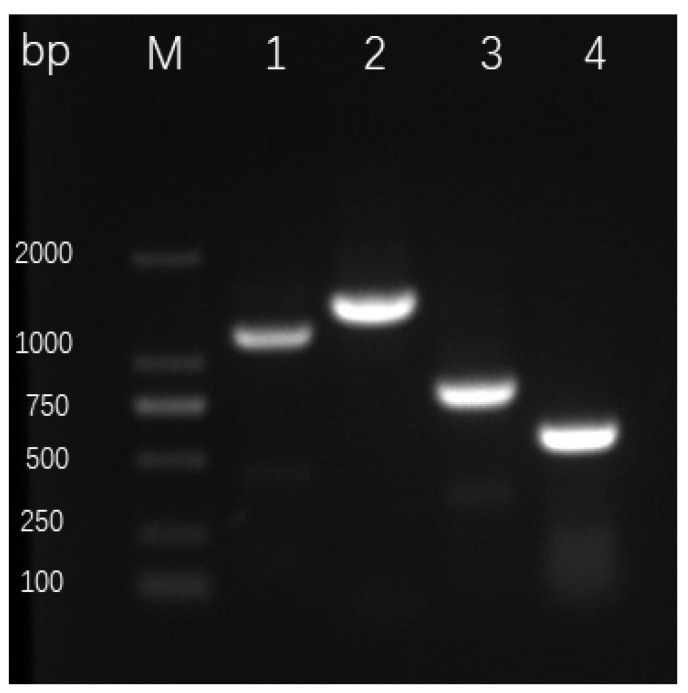
Detection of antibiotic genes in ST–TJ4 by PCR amplification (lane M, DNA marker; 1, PCA; 2, PHZ; 3, PRN; 4, PM).

**Figure 4 microorganisms-08-00590-f004:**
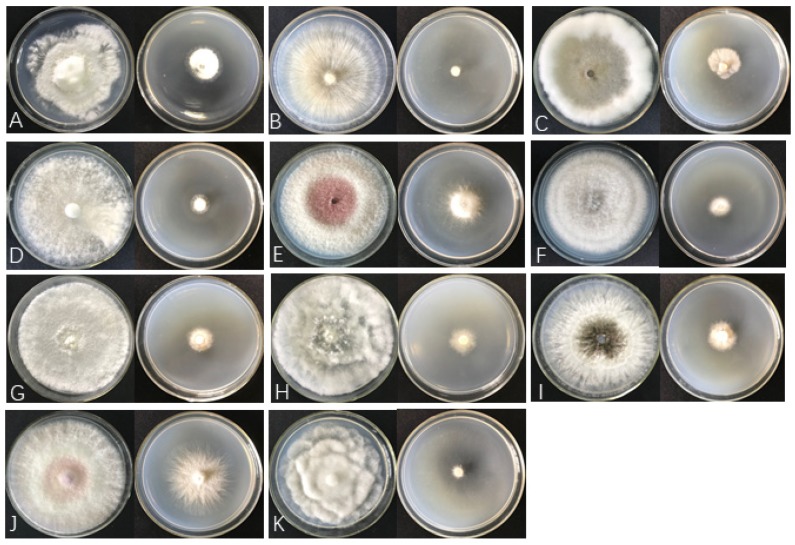
The antifungal spectrum of strain ST–TJ4 VOCs. Plate on the right treated with VOCs, plate on the left untreated. (**A**) *Pestalotiopsis versicolor*; (**B**) *Rhizoctonia solani*; (**C**) *Fusicoccus aesculi*; (**D**) *Cytospora chrysosperma*; (**E**) *Fusarium graminearum*; (**F**) *Colletotrichum tropicale*; (**G**) *Phomopsis ricinella*; (**H**) *Botryosphaeria berengeriana*; (**I**) *Sphaeropsis sapinea*; (**J**) *Fusarium oxysporum* and (**K**) *Phytophthora cinnamomi.*

**Figure 5 microorganisms-08-00590-f005:**
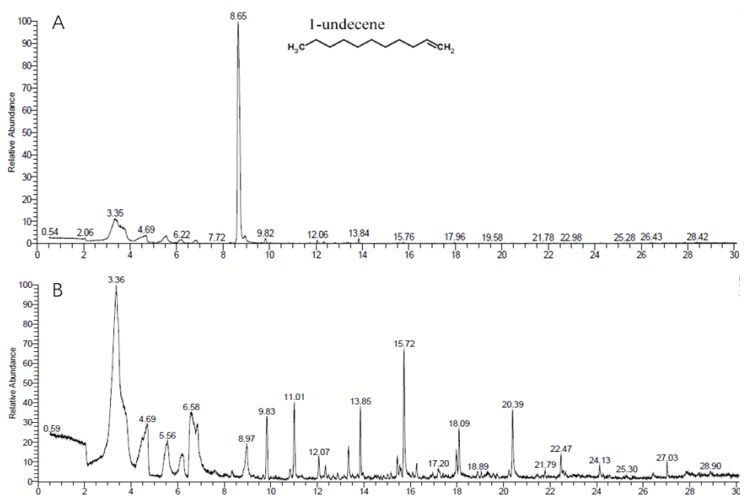
Chromatographic profiles of the volatile organic compounds (VOCs) of (**A**) strain ST–TJ4 incubated for 48 h in LB medium and (**B**) uninoculated LB medium.

**Figure 6 microorganisms-08-00590-f006:**
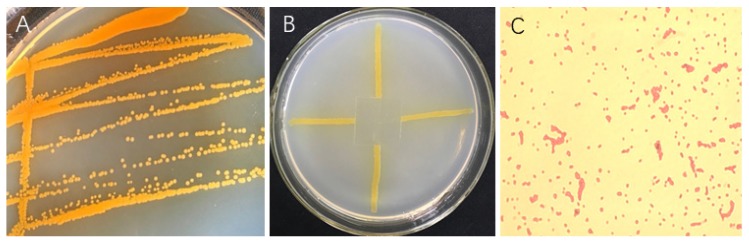
Morphological identification: colony morphology (**A**), oxygen demand test results (**B**), and Gram staining (**C**) of *Pseudomonas* sp. ST–TJ4.

**Figure 7 microorganisms-08-00590-f007:**
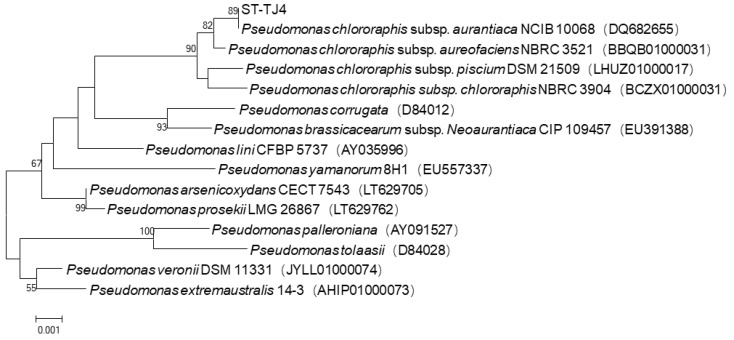
Phylogenetic tree derived from the 16S rRNA-encoding gene sequences of strain ST–TJ4 and related taxa of *Pseudomonas*. The tree was constructed using the neighbor-joining method using MEGA 7.0. The bootstrap test was conducted with 1000 replicates. Bootstrap values >50% were indicated at the nodes. The scale bar represents substitutions per nucleotide position.

**Table 1 microorganisms-08-00590-t001:** Oligonucleotide primers used in this study.

Primer	Sequence	Target	Antibiotic or Related Pathways	Size (bp)
pca2a	TTGCCAAGCCTCGCTCCAAC	phzCD	1-Phenazinecarboxylic acid	1150
pca3b	CCGCGTTGTTCCTCGTTCAT			
PHZ1	GGCGACATGGTCAACGG	phz	Phenazine biosynthesis	1408
PHZ2	CGGCTGGCGGCGTATTC			
PRNCF	CCACAAGCCCGGCCAGGAGC	prnC	Pyrrolnitrin	786
PRNCR	GAGAAGAGCGGGTCGATGAAGCC			
PM1	TGCGGCATGGGCGTGTGCCATTGCTGCCTGG	hcnAB	Hydrogen cyanide	570
PM2	CCGCTCTTGATCTGCAATTGCAGGCC			

**Table 2 microorganisms-08-00590-t002:** Mycelia growth of phytopathogenic fungi inhibited by strain ST-TJ4.

Target Pathogens	Percent Inhibition (%)	Mean ± SE
	Diffusible	Volatile
*Botryosphaeria berengeriana*	53.49 ± 4.8 ^cd^	73.76 ± 1.7 ^bcd^
*Colletotrichum tropicale*	21.46 ± 2.0 ^a^	86.25 ± 5.1 ^cd^
*Cytospora chrysosperma*	23.18 ± 6.9 ^ab^	82.79 ± 0.5 ^cd^
*Fusarium graminearum*	28.04 ± 8.4 ^ab^	55.62 ± 7.1 ^ab^
*Fusarium oxysporum*	41.11 ± 5.3 ^bcd^	40.4 ± 5.5 ^a^
*Fusicoccus aesculi*	35.98 ± 9.9 ^abc^	72.95 ± 13.38 ^bcd^
*Pestalotiopsis versicolor*	49.00 ± 4.9 ^cd^	62.31 ± 15.6 ^abc^
*Phomopsis ricinella*	59.91 ± 3.9 ^d^	73.03 ± 7.4 ^bcd^
*Phytophthora cinnamomi*	50.58 ± 5.2 ^cd^	91.35 ± 1.1 ^d^
*Rhizoctonia solani*	19.87 ± 4.4 ^a^	90.63 ± 0.4 ^d^
*Sphaeropsis sapinea*	56.73 ± 2.6 ^d^	71.97 ± 3.6 ^bcd^

In the same row data followed by the different, same, and overlapping lower-case letters means significantly different, and no significantly different of their overlapping to Duncan’s multiple range test at *p* < 0.01. Each result presents the mean ± standard derivation from three replicates.

**Table 3 microorganisms-08-00590-t003:** GC-MS/MS volatile profile of strain ST-TJ4.

Retention Time (min)	Relative Peak Area (%)	CAS#	Compound
2.04	1.42	5874-90-8	l-Ala-l-Ala-l-Ala
6.85	1.14	541-05-9	octamethylcyclotetrasiloxane
8.65	75.97	821-95-4	1-undecene
8.95	1.06	2078-13-9	4-hydroxybenzoic acid
12.04	0.54	53044-27-2	phosphonoacetic acid

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
