# Peer review of "Forest Tree Associated Bacterial Diffusible and Volatile Organic Compounds against Various Phytopathogenic Fungi"

_microorganisms, 2020, doi:10.3390/microorganisms8040590_

Round 1
Reviewer 1 Report
This manuscript describes the antibiotic effects of a bacterial isolate from the rhizosphere of poplar on a broad range of ascomycete, basidiomycete, and oomycete plant pathogens. The authors used biochemical, morphological, and molecular methods to identify the bacterium (strain ST-TJ4) as Pseudomonas chlororaphis subsp. aurantiaca. The authors demonstrate the antifungal effects through the production of diffusible compounds (plate inhibition test) as well as volatile compounds using a “sealed base plate” method, and they also identified some candidate compounds that may be responsible for this effect. The authors also demonstrated the presence of some previously identified antibiotic genes using PCR. The authors provide evidence of the possible mechanisms of inhibition of plant pathogens by this bacterial strain. Overall, the work is competently performed and clearly described. A particular strength of the work is the investigation of a variety of possible mechanisms of inhibition by the strain and the provision of evidence for these. In a few places, however, the evidence is rather weak and needs further support; this is particularly true of the strain identification using 16S rRNA-encoding gene sequencing, which is a notoriously low-resolution method particularly below the genus level. I have expanded on this and other points below.
Abstract
- Line 16 and elsewhere in the manuscript: “16S rDNA” does not exist. While this is a commonly used shorthand, it is incorrect. Bacterial ribosomes contain 16S rRNA, which is encoded by one or more genes in the genome. Please consider changing to, “16S rRNA-encoding gene” or something similar.
Methods
- Line 77: “we dipped in the overnight culture of ST-TJ4 bacterial suspension…”. This needs further clarity. I think the authors may mean that they dipped a sterile swab or loop into a bacterial suspension, then used that to streak the PDA plate to examine the effect of diffusible compounds produced by ST-TJ4 on the fungi. Please clarify.
- Lines 100-107: Please clarify the method of employed using two sealed base plates, particularly since no literature references are provided. A diagram may help here; it is not clear from the description exactly how these plates were set up. How were fungi exposed to the volatiles produced by the bacterium to produce the antifungal effects observed? It is not clear from this description. Please clarify for the benefit or reproducibility.
- Lines 113-114: flasks were sealed with tin foil to prevent volatile compounds from escaping? This hardly seems like an airtight seal. Even parafilm covered with tinfoil would have been better, but a rubber stopper probably would have been necessary to achieve this effect. How did the authors verify that they did not lose any volatile compounds with such a weak seal on the flasks? Please comment or clarify.
- Lines 115-118: please define the nonstandard acronyms used in this part of the description (eg PDMS/DVB, SPME).
Results
- Line 153: The authors mention that ST-TJ4 “significantly” inhibited the growth of the specified pathogens, but neither provided the statistical test used (unless that can be inferred from the Methods) nor provided the significance cutoff used. Please clarify the language used here for the benefit or readers.
- Lines 166-169 (Figure 2 legend): The authors describe this figure as showing the results of tests for fungal cell wall degrading enzymes, including cellulase using CMC plates. Cellulose is not present in fungal cell walls, so the positive results observed do not indicate the production of fungal cell wall degrading enzymes. Fungal cell walls are composed primarily of chitin along with some other polysaccharides, and interestingly ST-TJ4 was negative for the production of chitinase, which suggests that the direct lysis of fungal cells was not an important antifungal mechanism for this strain. Please comment and clarify.
- Figure 4: I found the names of the pathogens somewhat hard to read on this panel. It might be easier for readers if the authors just lettered the plates and provided the names in the legend below. Please consider clarifying.
- Table 2: While it is likely, readers are left guessing that the superscript letters indicate statistically significant differences. Please specify this as a Table footnote, and provide the statistical test used (even though it is mentioned in the methods, it is good to repeat here) and the significance cutoff (eg p <0.01).
- Line 216: This is an important typographical error: the authors indicate that strain ST-TJ4 shared 89% nucleotide identity at 16S, when it is 99%. That is a huge difference. Please correct. Moreover, when the 16S sequence of ST-TJ4 provided by the authors is compared to GenBank using a simple BLAST, a multitude of Pseudomonas species are identified with 99% or greater sequence identity with ST-TJ4, including strains of P. chlororaphis, P. brassicacearum, P. tolaasii, P. fluorescens, and others. This reflects the fact that 16S rRNA-encoding gene sequences are very typically unable to provide resolution below the genus level in many cases, and the sequence would certainly have a great deal of trouble below the species level (authors claim to identify the strain to the subspecies level). I am not convinced that the authors have provided sufficient evidence that ST-TJ4 indeed corresponds to the species/subspecies claimed. This would require more sequencing evidence from other taxonomic or biochemical markers. Please soften the claim (eg Pseudomonas sp. ST-TJ4), or provide further evidence of strain identity.
Discussion
- Line 239: again, the evidence presented in this manuscript is inadequate to identify Pseudomonas sp. ST-TJ4 even to the species level, let alone subspecies. Please clarify.
- Lines 259-260: As mentioned above, the production of cellulases by Pseudomonas sp. ST-TJ4 is irrelevant in the context described since fungal cell walls do not contain cellulose. Please correct the language around this.
- Line 269: “lower fungi” is an outdated term to describe oomycetes. Evidence clearly indicates that oomycetes are more closely related to algae and green plants compared to true Fungi, which are more closely related to animal cells. Please clarify.
Author Response
Point 1: Line 16 and elsewhere in the manuscript: “16S rDNA” does not exist. While this is a commonly used shorthand, it is incorrect. Bacterial ribosomes contain 16S rRNA, which is encoded by one or more genes in the genome. Please consider changing to, “16S rRNA-encoding gene” or something similar.
Response 1: Thanks for the referee’s suggestion, we have revised these expressions in the full manuscript. The revised details can be found in lines 16,22,25, page 1; line 142, page 4; line 239, page 10; line 142, page 4.
Point 2:Line 77: “we dipped in the overnight culture of ST-TJ4 bacterial suspension…”. This needs further clarity. I think the authors may mean that they dipped a sterile swab or loop into a bacterial suspension, then used that to streak the PDA plate to examine the effect of diffusible compounds produced by ST-TJ4 on the fungi. Please clarify.
Response 2: We are sorry for not addressing the methods clearly. We added this explanation in the revised manuscript and the details can be found in Line 78, Page 2.
Point 3:Lines 100-107: Please clarify the method of employed using two sealed base plates, particularly since no literature references are provided. A diagram may help here; it is not clear from the description exactly how these plates were set up. How were fungi exposed to the volatiles produced by the bacterium to produce the antifungal effects observed? It is not clear from this description. Please clarify for the benefit or reproducibility.
Response 3: According to the suggestions of the reviewer, we quoted relevant reference, the details can be found in Line 102, Page 3. The device can avoid direct physical contact between bacteria and fungi, only gas communication. A diagram will be uploaded to the manuscript submission system as supplementary Figure S1.
Point 4: Lines 113-114: flasks were sealed with tin foil to prevent volatile compounds from escaping? This hardly seems like an airtight seal. Even parafilm covered with tinfoil would have been better, but a rubber stopper probably would have been necessary to achieve this effect. How did the authors verify that they did not lose any volatile compounds with such a weak seal on the flasks? Please comment or clarify.
Response 4: Thank you for your concern. The relevant experimental steps and instrument settings referred to published journal papers with some modifications. Sealing with tin foil is beneficial to insert the extraction head into the conical bottle for the collection of VOCs.
Point 5: Lines 115-118: please define the nonstandard acronyms used in this part of the description (eg PDMS/DVB, SPME).
Response 5: According to the suggestion of the reviewer, we have defined the full name of PDMS/DVB. The revised details can be found in line 116, page 3. The full name of SPME has been explained in the manuscript and this can be found in line 111, page 3.
Point 6: Line 153: The authors mention that ST-TJ4 “significantly” inhibited the growth of the specified pathogens, but neither provided the statistical test used (unless that can be inferred from the Methods) nor provided the significance cutoff used. Please clarify the language used here for the benefit or readers.
Response 6: We are very sorry for our negligence. According to the agar-diffusible substances produced by ST-TJ4 strain, we analyzed the significant differences in the inhibitory effects of different fungi, which were shown in Table 2, but not mentioned in parentheses. We added this explanation in the revised manuscript and the details can be found in Line 156, Page 4.
Point 7: Lines 166-169 (Figure 2 legend): The authors describe this figure as showing the results of tests for fungal cell wall degrading enzymes, including cellulase using CMC plates. Cellulose is not present in fungal cell walls, so the positive results observed do not indicate the production of fungal cell wall degrading enzymes. Fungal cell walls are composed primarily of chitin along with some other polysaccharides, and interestingly ST-TJ4 was negative for the production of chitinase, which suggests that the direct lysis of fungal cells was not an important antifungal mechanism for this strain. Please comment and clarify.
Response 7: Chitin, glucan and protein are the main components of the cell wall of most fungi, while cellulose is the main component in the cell wall of a few oomycetes. Since Phytophthora camphora was included in the fungal spectrum selected by the author, the production of cellulase was also determined. And many studies have also reported Loss of turgidity and tearing of the pathogen hyphae may both be attributable to cellulase and protease activities(Chen Xiaomeng, et al. The plant pathology journal,2019. doi:10.5423/PPJ.OA.03.2019.0064). However, as the reviewer said, ST-TJ4 strains may take different antagonistic strategies against different pathogenic fungi. The next step is to select a pathogenic fungus to construct mutants of bacterial antifungal substances in order to clarify the role of various antagonistic substances in inhibiting the growth of pathogenic fungi.
Point 8: Figure 4: I found the names of the pathogens somewhat hard to read on this panel. It might be easier for readers if the authors just lettered the plates and provided the names in the legend below. Please consider clarifying.
Response 8: We have revised Figure 1 and Figure 4 based on the reviewer's comments.
Point 9: Table 2: While it is likely, readers are left guessing that the superscript letters indicate statistically significant differences. Please specify this as a Table footnote, and provide the statistical test used (even though it is mentioned in the methods, it is good to repeat here) and the significance cutoff (eg p <0.01).
Response 9: We apologize for not specifying the superscript letters clearly. We added this explanation in the revised manuscript and the details can be found in Lines 202-205, Page 8.
Point 10: Line 216: This is an important typographical error: the authors indicate that strain ST-TJ4 shared 89% nucleotide identity at 16S, when it is 99%. That is a huge difference. Please correct. Moreover, when the 16S sequence of ST-TJ4 provided by the authors is compared to GenBank using a simple BLAST, a multitude of Pseudomonas species are identified with 99% or greater sequence identity with ST-TJ4, including strains of P. chlororaphis, P. brassicacearum, P. tolaasii, P. fluorescens, and others. This reflects the fact that 16S rRNA-encoding gene sequences are very typically unable to provide resolution below the genus level in many cases, and the sequence would certainly have a great deal of trouble below the species level (authors claim to identify the strain to the subspecies level). I am not convinced that the authors have provided sufficient evidence that ST-TJ4 indeed corresponds to the species/subspecies claimed. This would require more sequencing evidence from other taxonomic or biochemical markers. Please soften the claim (eg Pseudomonas sp. ST-TJ4), or provide further evidence of strain identity.
Response 10: We are very sorry for our negligence. It is true that the 16s rRNA-encoding gene sequence can not accurately identify the species of bacteria. According to the suggestions of the reviewer, we have revised the relevant expressions in the manuscript. The revised details can be found in lines 227-229, page 9. Multi-gene tandem analysis and biochemical experiments will be carried out in next work.
Point 11: Line 239: again, the evidence presented in this manuscript is inadequate to identify Pseudomonas sp. ST-TJ4 even to the species level, let alone subspecies. Please clarify.
Response 11: Considering the Reviewer’s suggestion, this sentence has been revised (Line 253, page 10).
Point 12: Lines 259-260: As mentioned above, the production of cellulases by Pseudomonas sp. ST-TJ4 is irrelevant in the context described since fungal cell walls do not contain cellulose. Please correct the language around this.
Response 12: According to the suggestion of the reviewer, we corrected the language around this (Line 275, page 11).
Point 13: Line 269: “lower fungi” is an outdated term to describe oomycetes. Evidence clearly indicates that oomycetes are more closely related to algae and green plants compared to true Fungi, which are more closely related to animal cells. Please clarify.
Response 13: The reviewer's suggestion is pertinent, we revised the expression (Line 284, page 11).
Reviewer 2 Report
The authors have studied whether plant growth-promoting rhizobacteria can potentially be used as an alternative strategy to control plant diseases.
In my modest opinion, I think that it is an interesting approach to control plant diseases. The idea is nice to me, the methods are well presented and, as a result, I like the paper.
A minor amendment is in the pdf attached.

Author Response
Response to Reviewer 2 Comments
Point 1: ... and "something"? or "and" must be delete.
Response 1: Thank you very much for your careful discovery. We apologize for our negligence. The revised details can be found in Lines 41-42, page 1.